# Scale of Operation, Financial Support, and Agricultural Green Total Factor Productivity: Evidence from China

**DOI:** 10.3390/ijerph19159043

**Published:** 2022-07-25

**Authors:** Li Wang, Jinyang Tang, Mengqian Tang, Mengying Su, Lili Guo

**Affiliations:** 1College of Economics, Xihua University, Chengdu 610039, China; 0720050001@mail.xhu.edu.cn; 2College of Economics, Sichuan Agricultural University, Chengdu 611130, China; 201907298@stu.sicau.edu.cn (J.T.); tangmengqian@stu.sicau.edu.cn (M.T.); 3College of Economics, Guangxi Minzu University, Nanning 530006, China

**Keywords:** agricultural green total factor productivity, scale of operation, financial support, ARDL

## Abstract

Large-scale agricultural operations number among the ways to promote the green development of the agricultural sector, which can not only encourage farmers to adopt green innovative technology, reduce the input of chemical fertilizers and pesticides, and achieve environmental protection, but it also enables production with a high efficiency through an economy of scale and an improvement in farmers’ income. Based on the agricultural panel data of 30 provincial administrative regions in China from 2000 to 2019, the panel autoregressive distribution lag model was used to explore the dynamic relationship between a business’ scale, financial support, and agricultural green total factor productivity (AGTFP). The empirical outcomes indicate that there is a significant cross-sectional dependence, cointegration relationship, and long-run relationship between the scale of agricultural operations, financial support for agriculture, and AGTFP. Strengthening the intensity of financial support for agriculture is not conducive to improving AGTFP. On the contrary, increasing the scale of agricultural operations could promote AGTFP. In addition, the panel Granger causality test results indicate that financial support for agriculture has a unidirectional causal relationship with the scale of agricultural operations and AGTFP. The impulse response results demonstrate that reducing part of the financial support for agriculture or increasing the scale of operation can promote AGTFP. These conclusions have a long-term practical significance for agricultural departments and decision-making regarding financial distribution.

## 1. Introduction

Since its reform and opening-up in 1978, China’s economy has achieved rapid growth objectives and has become the second-largest economy and the largest developing country since 2010. China’s agriculture has undergone a comprehensive and in-depth reform that has enabled remarkable achievements and has made an essential contribution to the development of world agriculture [1]. However, with the rapid growth of the agricultural economy, China’s agricultural production has caused great environmental damage, and a large amount of greenhouse gases have been emitted, which are detrimental to the sustainable development of agriculture [2]. For example, from 2007 to 2017, China’s agricultural fertilizer use increased by 14.71%, from 51.078 million tons to 58.594 million tons. The extensive input of agricultural production factors has caused large amounts of greenhouse gas emissions and agricultural non-point source pollution [3,4]. Under the background of the current tight agricultural environment, only by adjusting the pattern of agricultural development and improving agricultural green total factor productivity (AGTFP) can we realize green sustainable development in Chinese agriculture [5]. To achieve this goal, the Chinese central government has formulated policies to promote green agriculture development that have been stated in the Central No.1 Document in 2022. At the same time, the government has achieved positive results regarding strengthening the science and technology-based support for green agriculture, subsidizing green agriculture, improving the utilization rate of production factors, and reducing pollution. The Chinese government has also attracted an increasing number of scholars to explore the green and sustainable development of agriculture.

Government subsidies to agriculture significantly reduce farmers’ use of chemical fertilizers and promote green development. However, some scholars put forward different views, arguing that financial subsidies do not always promote the green development of agriculture. For example, subsidies for agricultural output can promote agricultural overproduction and cause pollution [6,7]. In addition, although the increase in the scale of agricultural operations has generally played a positive role in promoting the use of green technology and improving production efficiency, showing a clear environmental protection effect [8,9], blindly expanding the scale of operation due to farmers’ pursuit of profits is not conducive to green agricultural production. Under such circumstances, it is very important to clarify the causal relationship between fiscal support for agriculture, the scale of agricultural operations, and AGTFP. For example, if the increase in the scale of agricultural operations promotes AGTFP, the state should formulate policies to guide traditional household-based agricultural production to develop in the direction of intensification. If the intensity of fiscal support for agriculture has a significant negative impact on AGTFP, relevant departments should consider how to correctly guide the flow of fiscal and monetary funds, rather than simply increasing the proportion of agriculture-related expenditures.

The primary purpose of this study is to use the agricultural panel data of 30 provincial administrative regions in China from 2000 to 2019 and employ ARDL and PVAR models to explore the relationship between AGTFP, the scale of agricultural operations, and financial support for agriculture. Firstly, the paper uses the SBM model, including bad outputs, to measure each province’s AGTFP. Secondly, this research uses the unit root, cointegration, Granger causality, and other methods to test the core variables of this paper. Thirdly, the paper studies the relationship between three variables by using the panel data model. Compared with the previous literature on agricultural production efficiency, this paper takes the carbon emissions from agricultural production into account when calculating the index of agricultural production efficiency. Since carbon emissions are the main focus of environmental protection and there are carbon emissions in all aspects of agricultural production, calculating agricultural carbon emissions can better reflect green production efficiency. In addition, the PVAR model proposed by Holtz-Eakinetal [10] has many advantages over the VAR model and panel data. The three variables of interest are regarded as endogenous variables, and the innovative shock of one endogenous variable on other endogenous variable is analyzed by calculating the orthonormal impulse response function. Based on the empirical results, the paper also suggests policy suggestions to promote green agricultural production and improve production efficiency.

Compared with the existing literature, this paper has the following three contributions: (1) this paper places AGTFP, the scale of agricultural operations, and the strength of the supporting agriculture into the integrated framework and uses the panel model to analyze the causal relationship between them, which further confirms the positive role of the scale of agricultural management in agricultural green development. At the same time, it is also observed that the overall effect of the strength of the supporting agriculture on the green development of agriculture is negative, which provides a foundation for more detailed research on the role of agricultural financial funds and the accurate guiding of financial funds in the positive direction. (2) The empirical method of this paper considers cross-sectional correlation, cointegration relationships, the lag effect, and a variety of other methods while mutually verifying the results to ensure their robustness. (3) The research of this paper can provide a reference for developing green agriculture and improving agricultural total factor green productivity. In addition, it will provide reasonable policy suggestions for agricultural scale and financial support.

## 2. Literature Review

Upon reviewing the literature, it is evident that many previous works have discussed the influencing factors of agricultural green total factor productivity (AGTFP). For example, taking the opening of high-speed rail systems as an instrumental variable, in the form of regression discontinuous design (RDD) and two-stage least squares (2SLS) estimation, Wang et al. [11] found that interregional investment can significantly affect AGTFP by affecting the number of green invention patents. To analyze each province’s green total factor productivity, Wang et al. [12] used stochastic frontier analysis (SFA) and Malmquist index methods. They found that the green technological innovation of the province can significantly improve its AGTFP but restrains the neighboring provinces’ AGTFP. Yuan and Zhang [13] analyzed the impact of environmental regulations on the input and output of production factors and found that environmental regulation has an inverted “U” curve on AGTFP. However, the allocation rate of production factors can reverse the inhibitory effect of strict environmental regulations on agricultural total factor productivity. Moreover, a mix of different regulatory policies and investing in human capital more positively affect the high-quality development of China’s agricultural economy. Using the three-stage Data Envelopment Analysis (DEA) method combined with the Slack-Based Measure (SBM) model, Chen et al. [14] discussed the spatial distribution and changing trend of AGTFP. The results show that China’s overall green total factor productivity is low and decreases from east to west. Considering the negative effects of carbon emissions, Liu et al. [1] calculated China’s AGTFP using the Super-SBM model and found that the difference was caused by different agricultural factor endowments and regional characteristics. In addition, the strengthening of investment in agricultural research and sustainable development, the development of cleaner agricultural production, and the expansion of the degree of the opening-up of agriculture to the outside world all play a positive role in improving agricultural total factor productivity and promoting balanced rural development. Still, factor market distortion and inefficient production scale inhibit the improvement of AGTFP [1,15,16]. However, there are few studies of the relationship between the intensity of financial support for agriculture and AGTFP. Based on the existing literature, this paper comprehensively studies the effects of the intensity of the financial support for agriculture and the scale of agricultural operations on AGTFP.

In the stable growth of agricultural production, financial support and agriculture-related public policies play an important role; for example, research and dissemination investment in agricultural chemical fertilizer can contribute to agricultural growth [17]. Moreover, improving financial incentives can encourage financial institutions to increase loans to agriculture, reduce the income gap between urban and rural areas, promote sustainable development, and increase green total factor productivity [18]. In addition, the government’s public financial investment in agricultural infrastructure, education, research, and so on can promote social capital investment in agriculture. However, other input subsidies, except for irrigation subsidies, do not achieve the goal of increasing social capital investment and promoting agricultural growth. Therefore, the government should eliminate some input subsidies and shift the resources to public agriculture investment [19]. Akbar and Jamil [17] also pointed out that the government should increase general infrastructure investment rather than public investment in the agricultural sector due to the crowding-out effect of the agricultural sector. In addition, increasing credit support for agriculture and financial support for policy promotion can promote the use of green agricultural technologies and the development of green agriculture [20,21,22]. Whether China’s financial support for agriculture can promote AGTFP is of vital importance to the healthy development of agriculture, the reduction of environmental pollution, and the livelihoods of developing countries [12,17].

The sustainable development of agriculture can be simply summarized as meeting people’s current needs for agricultural products without harming the ability of future generations to develop agriculture following the Brundtland Report (1987). However, due to the different evaluation systems, the interpretation of agricultural sustainable development is vague and complex [23,24]. Agricultural green total factor productivity (AGTFP) reveals a facet of steady growth that exceeds input factors under environmental pressure, and it is an accurate indicator of agricultural economic performance combined with the ecological environment [14]. Compared with agricultural total factor productivity, AGTFP takes into account environmental constraints, reflecting economic growth and environmental protection [25]. The main way to improve AGTFP and achieve the goal of agricultural sustainable development is to promote green technological innovations [12]. The innovation and application of agricultural green technology are strongly related to the scale of agricultural operations. Agricultural producers with a larger operating scale are more willing to use new technology and spend more time and money pursuing agricultural knowledge [8,26]. Furthermore, from the regional aspect, the regional green agricultural technology-related progress is higher in the generally larger scale of agricultural management [26]. The scale of agricultural operations plays an essential role in adopting new technology. Many studies show a significant positive relationship between the scale of agricultural operations and agriculture-related green technology’s progress [26,27]. Their theoretical framework is as follows: unlike the smaller scale of agricultural operation, farmers operating at a larger scale can try to use innovative green technology on their portion of their land [28]. In addition, the use of some green innovative technologies requires the support of economies of scale [29]. The relationship between agriculture and the environment is inseparable and the two facets circularly affect each other [30]. From the point of view of reducing environmental pollution, the scale of agricultural operations has a significant negative influence on the intensity of pesticide use. On average, every 1% increase in farm size reduces the pesticide used per hectare by 0.2%. Therefore, measures to improve large-scale agricultural management can promote the development of green agriculture [9,31].

The appropriate scale of agricultural operation is the core factor of sustainable development. In most agricultural countries, farm-scale growth is the key to rapid economic growth, poverty reduction, and stable rural development [32]. However, it has always been controversial whether the larger or smaller scale of agricultural operations should be adopted. Some studies have found that the smaller scale of agricultural operations has a higher land yield per hectare than the larger. That is, there is an inverse correlation between the scale of agricultural operations and agricultural productivity [33,34]. However, some studies have pointed out that the persistence of a smaller the scale of agricultural operations may restrict overall agricultural growth and competition. With the growth of the economy and market, a smaller scale of agricultural operations will develop in the direction of scale reward [35,36]. From the point of view of China and from a global perspective, Ren et al. [9] found that the expansion in the scale of agricultural operations has a positive role on farmers’ net profit, economy, technology, and labor efficiency. However, the relationship between the scale of agricultural operations and AGTFP is not still clearly understood.

## 3. Data Source and Description

### 3.1. The Scale of Agricultural Operations

In previous studies, the measurement of the scale of agricultural operations has been shown to differ between macro and micro studies. From a micro point of view, the scale of agricultural operations is expressed by the land size of each farm [26,30,31,37]; from a macro perspective, the scale of agricultural operation is expressed by the per capita cultivated land area of each region or the sown area of each agricultural laborer [26,31]. Due to historical reasons for agricultural production, China has still been dominated by small-scale peasant production for the past 20 years, but it is also transforming some operations into forms of large-scale farm production [38,39]. Based on the above practical reasons, it is better to explore the impact of per capita cultivated land or per capita sown area on the agricultural green total factor productivity from a macro perspective than to use micro-farm data. Considering that the agricultural sown area is the actual land input of agricultural production, and the employed population of the primary industry is the human resource input of agricultural production, the agricultural operating area of this paper is represented by the per capita agricultural sown area. The value is the ratio of each province’s total agricultural sown area to the number of people engaged in the primary industry. To ensure the availability of data and the consistency of this paper’s statistical calibre, the original data of agricultural operating areas have been sourced from the China Rural Statistical Yearbook, including panel data of 30 provincial administrative regions from 2000 to 2019.

### 3.2. The Intensity of Financial Support for Agriculture

Previous studies have generally used the absolute amount of governmental expenditure related to agriculture, such as government subsidies for agricultural irrigation and agricultural research expenditure or per capita fiscal spending in agriculture, when calculating agriculture-related fiscal expenditure [19,40]. This type of analysis does not take into account the relative changes in agriculture-related fiscal expenditure. To study the influence of the change in financial support for agriculture on AGTFP, this paper uses the ratio of agriculture-related expenditure to the total financial expenditure to express the intensity of financial support for agriculture. The original data on the financial support for agriculture comes from China’s National Bureau of Statistics and China’s Rural Statistical Yearbook, including panel data from 30 provincial administrative regions from 2000 to 2019.

#### 3.2.1. Agricultural Green Total Factor Productivity (AGTFP)

Stochastic frontier analysis (SFA) and data envelopment analysis (DEA) are mainly used in the literature on calculating green total factor productivity. Wang et al. (2021) [12] used SFA and the Malmquist index method to estimate the green total factor productivity of provinces in China. However, few researchers use the SFA method to study AGTFP because the SFA method must set the probability distribution from a random error term; additionally, the frontier production function is affected by individual regions [41,42]. Compared with SFA, as a nonparametric method, DEA does not need to set specific production functions and inefficiency items in advance and is almost free from subjective influence [14]. Therefore, researchers regard data processed through DEA as the main measure of green total factor productivity [1]. Considering that the DEA model has shown its reliability and effectiveness in the extensive research of AGTFP [1,14,15] and the characteristics of multi-input and multi-output in agricultural production, this paper uses the DEA model. Since agricultural production will produce a variety of pollutants such as carbon emissions, we use an unexpected output as part of the output variable to test agricultural production efficiency, which is called agricultural total factor productivity, denoted herein as AGTFP. The Undesirable Output Model in the application DEA-Slover-Pro has two variants: Bad-Output and Non-Separable models. This Bad-output model can deal with expected and unexpected outputs. This paper uses the Bad-output model to measure AGTFP. The Bad-output model based on the SBM model modification is a special SBM model with an undesirable output, and based on the variable returns to the scale, the process of using the Undesirable Output model can be expressed as follows.
(1)$$P=\left\{\left(x,y^{g},y^{b}\right)|x\geq X\lambda ,y^{g}\leq Y^{g},y^{b}\leq Y^{b},L<e\lambda <U,\lambda \geq 0\right\}$$
L and U are the lower and upper bounds of the intensity vector, respectively, and the default of L and U is 1 in Dea-solver-pro. λ is the intensity vector. To allow the efficiency of every decision-making unit $\left(DMU\left(x_{0},y^{g},y^{b}\right)\right)$ to obtain the best frontier under the presence of an undesirable output variable, the input and output data must fulfill the requirements: $x_{0}\geq x,y_{0}^{g}\leq y^{g},y_{0}^{b}\geq y^{b}$
and
$\left(x,y^{g},y^{b}\right)\in P$. The Undesirable Output Model (or SBM model with an undesirable output) formula is as follows:(2)$$\rho =\min\frac{1-\frac{1}{m}\sum_{i=1}^{m}\frac{s_{i}^{-}}{x_{i0}}}{1+\frac{1}{s_{1}+s_{2}}\left(\sum_{r=1}^{s_{1}}\frac{s_{r}^{g}}{y_{r0}^{g}}+\sum_{r=1}^{s_{2}}\frac{s_{r}^{b}}{y_{r0}^{b}}\right)}\;s.t.\;x_{0}=X\lambda +s^{-},\;\;y_{0}^{g}=Y^{g}\lambda -s^{g},\;y_{0}^{b}=Y^{b}+s^{b},\;L\leq e\lambda \leq U\;s^{-}\geq 0,\;s^{g}\geq 0,\;s^{b}\geq 0,\;\lambda \geq 0$$
where vector
$\;s^{-}$, $s^{b}$, and $s^{g}$ indicate the excess quantities of input, undesirable output, and insufficient desirable output of the decision-making units, respectively. $s^{1}$ and $s^{2}$ represent the number of elements in $s^{b}$ and $s^{g}$. In the presence of a bad output, if and only if $\rho =0,s^{-}=0,s^{b}=0,s^{g}=0$, then the production efficiency can reach the optimal frontier domains. However, if 0<ρ<1, the efficiency of the decision-making unit is inefficient.

In terms of the input–output model, in this paper, the expected output is expressed by the total agricultural output value after the reduction of the agricultural gross output value index. The unexpected output is expressed by the carbon emissions generated by various substances. The calculation process of carbon emissions from agricultural production will be described in detail later on. Figure 1 shows the distribution of AGTFP in each province.

#### 3.2.2. Calculation of Agricultural Carbon Emission

With the continuous increase of China’s agricultural input, carbon emissions are increasing year by year, leading to a significant decline in environmental quality, which has become the main obstacle to the development of an agricultural green transformation [14]. There are many studies on carbon emission calculation, and the results obtained by different research methods may be different [1]. In this paper, according to the carbon source and carbon emission coefficient of agricultural production, the measurement formula of carbon emissions from agricultural production is as follows:(3)$$E=\sum E_{i}=\sum T_{i}\cdot \delta_{i}$$
where E represents the total carbon emission from agricultural production and $E_{i}$ is the carbon emission of a specific carbon variable. The carbon source variables used in this paper mainly include effective tillage area, agricultural chemical fertilizers and pesticides, agricultural plastic film, irrigated area, agricultural diesel fuel consumption, rural electricity consumption, and the feces from major livestock, pigs, cattle, and sheep. The source of all the above agricultural input data is China Rural Statistical Yearbook. $T_{i}$ is the amount of input variables. The carbon emission coefficient δ is shown in Table 1 [43].

### 3.3. Descriptive Statistics Analysis

Table 2 shows the summary data of various carbon sources needed to calculate the total carbon emissions required for AGTFP. According to the calculation, the value of AGTFP, the most critical variable in this study, is between 0.071 and 1, with an average value of 0.718 and a standard deviation of 0.166. Among the carbon sources required for calculation, the average consumption of agricultural chemical fertilizer, pesticide, agricultural plastic film, and agricultural diesel oil is 1,751,310 tons, 52,660 tons, 69,555 tons, and 638,270 tons, respectively; the average values of actual cultivated land area and effective irrigated area are 5.327 million hm2 and 2.004 million hm2, respectively; at the end of the year, the average number of cattle, pigs, and sheep in each region was about 357, 1516, 973 ten thousand respectively, and the final calculated total carbon emissions of each province averaged 6.3 million tons. The table also lists the data of the other two important variables studied in this paper, namely, the scale of agricultural operations (AGRISCALE) and the financial strength of the supporting agriculture (AGRIRATIO), with an average scale of operation of 0.602 hm2 per capita and an average fiscal expenditure ratio of about 0.092 per year. In order to computationally follow-up the model calculation and reduce endogenous problems, these three key variables were processed logarithmically. Next, we used the panel data of the scale of agricultural operations, the intensity of financial support for agriculture, and the AGTFP in 30 provincial administrative regions of China from 2000 to 2019 to analyze the relationship among them. Compared with cross-sectional and time-series data, panel data can provide large samples and reduce multicollinearity among variables, so it performs better in solving the problem of missing variables, providing more details about individual dynamic behaviors and improving the accuracy of parameter estimation [46].

## 4. Methodology

### 4.1. Test for Cross-Sectional Dependence

Since the agricultural time-series data of different provinces may be affected by the same impact, such as the change of the national unified agricultural policy, there is dependence among these provinces. Common impacts often lead to the dependence of elements in the panel, even though their influences may be inconsistent in the cross-sectional elements [47]. Cross-sectional correlation is the most important in panel data, and ignoring cross-sectional correlation often leads to inconsistent estimation and misleading information [48]. This paper uses the Breusch–Pagan LM test proposed by Breusch and Pagan (1980) [49] to check the robustness of our empirical results. We also used Pesaran CD proposed by Pesaran (2004) [50] and standardized La Grange Multiplier (LM) tests. The Pesaran CD test is based on the mean pairwise correlation coefficients of the OLS residuals derived from the standard augmented Dickey–Fuller (1979, ADF) [51] regression for each sequence [50,52]. The calculation formulas of the three cross-section inspection methods are as follows:(4)$$LM_{Breusch-Pagan}=\sum_{i=1}^{N-1}\sum_{j=i+1}^{N}T_{ij}\mu_{ij}^{2}\to \chi^{2}\left(\frac{N\left(N-1\right)}{2}\right)$$
(5)$$LM_{Pesaran}=\sqrt{\frac{1}{N\left(N-1\right)}}\sum_{i=1}^{N-1}\sum_{j=i+1}^{N}\left(T_{ij}\mu_{ij}^{2}-1\right)\to N\left(0,1\right)$$
(6)$$CD_{Pesaran}=\sqrt{\frac{2}{N\left(N-1\right)}}\sum_{i=1}^{N-1}\sum_{j=i+1}^{N}T_{ij}\mu_{ij}^{2}\to N\left(0,1\right)$$
(7)$$\mu_{ij}=\mu_{ji}=\frac{\sum_{t-1}^{T}\varepsilon_{\mathrm{ij}}\varepsilon_{ji}}{{\left(\sum_{t-1}^{T}\varepsilon_{\mathrm{ij}}^{2}\right)}^{\frac{1}{2}}\left(\sum_{t-1}^{T}\varepsilon_{\mathrm{jt}}^{2}{)}^{\frac{1}{2}}\right)}$$

The Breusch–Pagan LM test is suitable for panel data of small sizes and time *T*. Its formula is Equation (4). The cross-sectional tests proposed by Pasaran are suitable for panel data with large *N* and time *T*. Equation (5) is suitable for large size and changeable time *T*, and Equation (6) is suitable for large *N* and fixed *T*. $\mu_{ij}^{2}$ is the correlation coefficients obtained from the residuals of the Formula (6), where $\varepsilon_{\mathrm{ij}}$ and $\varepsilon_{ji}$ are standard errors [53].

### 4.2. Panel Unit Root Tests

Since the presence of horizontal unit roots in panel data may produce spurious regressions when using ordinary least squares (OLS) estimation techniques, a variable unit root test must be performed [25]. Before the cointegration analysis, it is necessary to determine that the variable is a first-order single integral. That is, the first-order difference is stationary [54]. According to previous studies, this paper uses four methods to test the panel unit root, involving the Levin–Lin–Chu test (LLC), the Im–Pesaran–Shin (IPS), Fisher’s augmented Dickey–Fuller (ADF) test, and Fisher’s Phillip and Perron (PP) test. Breitung (1999) [55] found that IPS suffers a dramatic loss of power when including individual trends and that the test is sensitive to the specification of deterministic trends [56], so the Breitung t-stat method is used in addition to the above four methods when considering intercept and trend terms.

### 4.3. Panel Cointegration Test

After the unit root tests, the paper uses Kao’s method (1999) to test the dynamic cointegration relationship between variables to determine whether the variables have a long-term relationship with economic norms. Kao (1999) extends the Engle–Granger two-step residual-based cointegration tests, especially in the first stage of estimation, requiring a cross-section-specific intercept and homogeneous coefficients on regression variables [57]. Its main formulas are as follows:(8)$$y_{it}=\alpha_{i}+\beta_{1}x_{1i}+\beta_{2}x_{2i}+\beta_{m}x_{mi}+e_{it}$$
(9)$$y_{it-1}+v_{it}=\alpha_{i}+\beta \left(x_{it-1}+\varepsilon_{it}\right)+e_{it}$$

In Equations (8) and (9), t is time series and i is a sample unit. $\alpha_{i}$ represent the individual intercept parameters, and $e_{it}\;$
are the residuals. Equation (8) is the first stage regression, and in the bivariate circumstance running Equation (9). The Kao (1999) approach is performed to test whether the residual,
$e_{it}$, derived from Equations (8) and (9) has a unit root.
(10)$$e_{it}=\rho_{i}e_{it-1}+v_{it}$$

With the null hypothesis of no cointegration, Kao establishes these statistics:(11)$$K_{p}=\frac{T\sqrt{N}\left(\rho -1\right)+3\sqrt{N}}{\sqrt{10.2}}$$
(12)$$K_{t}=\sqrt{1.25}t_{\rho }+\sqrt{1.875N}$$
(13)$$\mathrm{However},\;\rho >0:K_{ADF}=\frac{t_{\rho }+\sqrt{6N}\delta_{v}/\left(2\delta_{0v}\right)}{\sqrt{\delta_{0v}^{2}/\left(2\delta_{v}^{2}\right)+3\delta_{v}^{2}/(10\delta_{0v}^{2}})}$$

### 4.4. Granger Causality Test

If there is a long-term relationship between variables, it means that there is a causal relationship in at least one direction, which was proposed by Granger (2003). According to Granger (1969) [58], measuring the correlation between variables is not enough to fully understand the relationship between variables. The possible reason is that some correlations may be false and useless because there may be a third unexplained variable. Moreover, the correlation alone cannot judge the causal relationship between variables. So, knowing that our sequence is cointegrated, we must cross-check causality [59]. The paper uses the panel data Granger causality test extended by Dumitrescu and Hurlin (2012). The formulas are as follows:(14)$$y_{i,t}=\alpha_{0,i}+\alpha_{1,i}y_{i,t-1}+\ldots +\alpha_{k,i}y_{i,t-k}+\beta_{1,i}x_{i,t-1}+\ldots +\beta_{k,i}x_{i,t-k}+\varepsilon_{i,t}$$
(15)$$x_{i,t}=\alpha_{0,i}+\alpha_{1,i}x_{i,t-1}+\ldots +\alpha_{k,i}x_{i,t-k}+\beta_{1,i}y_{i,t-1}+\ldots +\beta_{k,i}y_{i,t-k}+\varepsilon_{i,t}$$
in which $y_{i,t}$ and $x_{i,t}$ measure the observed value of individual i in t periods. k represents the lag number of individuals; the coefficient of the individual is allowed to be different, but the hypothetical coefficient does not change with time. We used Eviews to test the Granger causality of panel data.

### 4.5. Autoregressive Distributed Lag

The autoregressive distributed lag model was created and estimated by Pesaran et al. [60,61]. The main advantage of the ARDL model is that it can deal with variables with different lag orders and analyze the well-known model similar to statistical regression [62]. In addition, the model can use relatively small data sets [46,63]. For long-run variable associations, we developed the following approach based on ARDL:(16)$$\begin{array}{cc} L_{AGTFP}=\alpha_{0}+ & \sum_{i=1}^{r}\sigma_{1}L_{AGTFP}{}_{t-i}+\\ & \sum_{i=1}^{r}\sigma_{2}L_{AGRIRITIO}{}_{t-i}+\sum_{i=1}^{r}\sigma_{3}L_{AGRISCALE}{}_{t-i}+\varepsilon_{t} \end{array}$$

In the equation mentioned above, $\sigma_{1},\sigma_{2},\mathrm{and}\;\sigma_{3}$ represent the long-run variance of the variables. The Akaike information criteria (AIC) were used to determine the correct lag. The following error correction model was employed for the short-run variables’ associations based on the ARDL model.
(17)$$\begin{array}{cc} L_{AGTFP}=\alpha_{0}+\sum_{i=1}^{r}\lambda_{1}L_{AGTFP}{}_{t-i} & +\sum_{i=1}^{r}\lambda_{2}{L_{AGRIRITIO}}_{t-i}+\\ & \sum_{i=1}^{r}\lambda_{3}L_{AGRISCALE}{}_{t-i}+\emptyset ECT_{t-i}+\varepsilon_{t} \end{array}$$

In the equation mentioned above, $\lambda_{1},\lambda_{2},\lambda_{3}$ represent the short-run changes. The error correction term (ECT) represents the short-term variance and measures the acceleration of change caused by fluctuations [64]. The error correction factor is expected to have negative and statistically significant coefficients, indicating that each shock is compensated at the next stage [65].

### 4.6. Impulse Response Approach and Variance Decomposition

Impulse response and variance decomposition can effectively test the impact of shocks on the research variables at present and in the future. They can be applied not only to the variables themselves but can also be transferred to other variables through the dynamic structure of the model. We used impulse response and variance decomposition to analyze the impact of the shock on the variables [53]. The estimation of impulse response appears in the stable adjoint matrix of the vector autoregressive (VAR) model [66], which can be written in the following form:(18)$$y_{t}=\sum_{j=0}^{k}\phi_{i}y_{t-i}+\varepsilon_{t}$$
(19)$$\phi_{i}=\{\begin{array}{c} I_{k}\;,\;i=0\\ \sum_{\dot{J}=1}^{i}\phi_{t-j}A_{j},\;i=1,2,\ldots \end{array}$$
(20)$$y_{it+h}-E\left[y_{it+h}\right]=\sum_{i=0}^{h-1}\varepsilon_{i}\left(t+h-1\right)\emptyset_{i}$$
(21)$$\sum_{i=0}^{hI}\theta_{nm}^{2}=\sum_{i=0}^{hI}{\left({i^{\prime }}_{m}K\emptyset_{in}\right)}^{2}$$
where the $\phi_{i}$ in (18) is an impulse response function, which can be estimated by moving the regression estimate using the infinite vector converted to (19). $y_{it+h}$ is the vector of the variable after period
 h in the t period(s), and $E\left[y_{it+h}\right]$ is its predicted value after period h in the t period(s). Equation (20) represents the contribution of variable n to the prediction error variance of variable m in the advance h period. $I_{k}$ is the unit element of the adjoint matrix,$\;i_{m}$ is the m column of $I_{k}$, $A_{j}$ is a coefficient matrix that transforms VAR into an infinite vector moving average, k is the optimal lag term, and $\varepsilon_{t}$ is the error term.

## 5. Results and Analysis

### 5.1. Cross-Sectional Dependence Tests Results

Table 3 shows the results obtained using three different cross-sectional correlation tests, all of which reject the zero hypothesis that there is no cross-sectional dependence at a 1% significance level.

### 5.2. Panel unit Root Tests Results

Table 4 shows the results of the five LLC, IPS, ADF, PP, and Breitung t-stat unit root test methods, including two types (only constant term, constant term, and trend term). Among them, the level series of the scale of agricultural operations is unstable in all tests; under the condition of the only constant term, the level series of the intensity of financial support for agriculture is a stationary series with a 1% significant level tested by three LLC, IPS, ADF methods. In contrast, the result tested by the PP method is unstable. When the level series of AGTFP includes a constant term and a trend term, it is tested by the LLC, ADF, and the PP method that it is a stationary series with a 1% significant level, but the result of the PP and Breitung t-stat approach is unstable. However, the first-order difference of all the variables is tested by either method to reject the null hypothesis that it is unstable, i.e., that it is a stationary series. Therefore, the unit root order of all the variables is within the first order or level series, which makes it possible to test the long-term trend between variables with Kao (1999).

### 5.3. Cointegration Tests and Causality Tests Results

The cointegration test results of AGTFP, financial support to agriculture, and the scale of agricultural management are shown in Table 5. At a significant level of 5%, Kao’s (1999) test statistics reject the null hypothesis that there is no cointegration relationship and support the alternative hypothesis that there is a cointegration relationship in the panel. The results confirm a long-term equilibrium causal relationship between the intensity of financial support for agriculture and the scale of agricultural operation towards AGTFP from 2000 to 2019. This will help to explore the previous goals of this paper and further examine the impact of financial support for agriculture and the scale of agricultural operations on AGTFP. As shown in Table 6, the intensity of financial support for agriculture has a unidirectional causal relationship to AGTFP and the scale of agricultural operations is significant at a level of 1%. This finding implies that the intensity of financial support for agriculture may affect AGTFP and the scale of agricultural operation. In addition, we cannot reject the null hypothesis between the agricultural green total factor and the scale of agricultural operations. That is, there is no causal relationship between AGTFP and the scale of agricultural operations. This implies no obvious time sequence between the scale of agricultural operations and AGTFP.

### 5.4. Autoregressive Distributed Lag results

The Akaike information criteria (AIC) have been used to determine the optimal lag length of the model. Finally, the model, after determining the parameters, yields ARDL (1,1,1). Table 7 shows the results of the long-term and short-term relationships between variables. The long-term relationship between variables shows that the coefficient of the scale of agricultural operations is 0.576, which is significant at the 1% level, and means that for every 1% increase in the scale of agricultural operations, the agricultural total factor productivity can increase by 0.576%. This result is in agreement with wang et al. (2015) [67] on the scale and productivity of rice farms in China. In addition, it can be explained by the fact that the increase in the scale of agricultural operations could reduce the use of pesticides, make it easier for large-scale agricultural producers to use green technology, enable more efficient production for large-scale agricultural producers, etc. [8,9,26,31]. Furthermore, there is no single economically optimal scale regarding agricultural operations, so policymakers need to rationally adjust the scale of agricultural operations according to the development of the economy and market [36]. The ARDL results further inmplicate that expanding the scale of agricultural operations is more suitable for China’s current economic development from the perspective of promoting green production. In addition, the coefficient of financial support for agriculture is −0.034, which is significant at the 1% level, indicating that for every 1% increase in financial support for agriculture, agricultural total factor productivity will decrease by −0.034%. Although this result is significant and consistent with the previous Granger causality test, its coefficient is very small. This situation is consistent with the chaotic conclusions drawn by the existing literature on the study of financial support for agricultural green development. That is, financial support for agriculture does not only have a positive effect on AGTFP by reducing the use of chemical fertilizers but also has a negative effect by subsidizing the output of agricultural products, which causes pollution through excessive production [6,7]. Therefore, this small negative coefficient is the total impact of fiscal agriculture-related expenditure on AGTFP after neutralizing the positive and negative effects. In addition, the table also shows the results of the short-term relationship between variables, and the coefficients of the scale of agricultural operations and financial support for agriculture are not significant. Therefore, we can say that the scale of the agricultural operation and the intensity of financial support for agriculture have no impact on AGTFP in the short term.

### 5.5. Impulse Response and Variance Decomposition Results

We needed to determine the best lag order of variables before using the VAR system to treat AGTFP, the intensity of financial support for agriculture, and the scale of agricultural operations as endogenous variables for impulse effect and variance decomposition analysis. The paper uses five methods involving the sequentially modified LR test statistic (LR), Final prediction error (FPE), Akaike information criterion (AIC), Schwarz information criterion (SIC), and Hannan–Quinn information criterion (HQ) to comprehensively judge the lag order of the optimal of the selected variables. As shown in Table 8, the optimal lag order is 3, and Figure 2 was obtained according to this order. It can be seen that each root is contained in the unit circle, which is in accordance with the condition of using the pulse effect and variance decomposition analysis based on the VAR model.

As can be seen from Figure 3, the variations in variables include not only innovative shocks of the variable itself but also the innovative shocks of other variables. In 15 forecast periods, 95.3% of the variation in AGTFP can be explained by innovative shocks themselves, 0.54% by innovative shocks of the scale of agricultural operations, and the remaining 4.15% by the intensity of the financial support for agriculture. This result accords with the causality test mentioned above, and the relationship between the intensity of financial support for agriculture and AGTFP is more significant. Furthermore, during these 15 forecast periods, the degree of explanation of the innovative shocks of AGTFP on the variation itself decreased, which means that the contribution of other driving forces increased. However, innovative shocks to the scale of the agricultural operation and the financial strength of the supporting agriculture on AGTFP have experienced a process of an initial rise and thena decrease. The innovative shocks of the financial support for agriculture are negative for most of the period, which implies that reducing the proportion of agriculture-related fiscal expenditure is conducive to improving AGTFP, and is thereby consistent with the results of the ARDL model. This finding shows that the influence of financial support for agriculture and the scale of agricultural operations has a lag effect on AGTFP, and the influence weakens with time after reaching the peak. In addition, innovative shocks to the intensity of financial support for agriculture can explain 7.2% of the variation in the scale of agricultural operations, and the innovative shocks of AGTFP can explain 0.72% of the variation in the scale of agricultural operations and 3.01% of the variation of financial support for agriculture.

### 5.6. Discussion

Due to the different research perspectives and the choice of indicators to measure the high-quality development of agriculture, previous studies obtained results that are different from this paper. Although the empirical test results of this paper indicate that the overall impact of the intensity of financial support for agriculture on AGTFP is negative, it may have a positive effect on AGTFP when specific to a certain type of specific fiscal expenditure. For example, Deng et al. (2021) found that public agricultural R&D investment is the main driver of China’s AGTFP improvement [68]; in Malawi, government subsidies for production factors for improving the efficiency of small-scale agricultural production are conducive to agricultural green development [69]. Within the provincial administrative regions, the impact of the scale of agricultural operations on AGTFP is negative. Hu et al. (2019) found that the smaller the scale of agricultural operation, the higher the utilization rate of the input factors in Jiangsu, China [70]. In addition, this paper mainly discusses the influence of the intensity of financial support for agriculture on AGTFP from a macro point of view, but this study still has some limitations, such as ignoring the role of the specific methods of financial support for agriculture and further research is required to explain the overall negative impact of the intensity of financial support for agriculture

## 6. Conclusions 

The main purpose of this study was to investigate the impact of the scale of agricultural operations and financial support for agriculture on agricultural green total factor productivity (AGTFP). Based on the agricultural panel data of 30 provincial administrative regions in China from 2000 to 2019, the SBM model, including the unexpected agricultural output, named carbon emissions, was used to calculate each province’s AGTFP. Then, the ARDL and impulse response method based on the VAR model were applied to conduct an empirical study on the relationship between variables. The empirical results are as follows. Firstly, the data after the first-order difference of the three variables are all stationary series, and there is a significant cross-sectional relationship and cointegration relationship among the three variables, demonstrating that the variables may be subject to the same impact in different provinces and may have a long-term and short-term equilibrium relationship. The Granger causality test showed that there is a one-way causal relationship between the financial support to agriculture and the scale of agricultural management. Secondly, the long-term and short-term relationship between variables using the ARDL model reveals the fact that the long-term effect of the scale of agricultural operations has a significant positive effect on AGTFP in China. In contrast, the long-term effect of financial support for agriculture suppresses the improvement of AGTFP, which is consistent with the pulse effect with the three order lag. Finally, the scale of the agricultural operation and the intensity of financial support for agriculture show no significant impact on AGTFP in the short term.

## 7. Policy Implications

Four following policy recommendations are put forward in accordance with the results above: (1) increase the agricultural land conversion rate and promote agricultural land agglomeration to form large-scale agricultural management units; (2) stop or limit subsidies for agricultural supplies with high carbon and pollution emissions, as doing so could not only effectively reduce the agricultural scale of non-clean production to pursue profits but also promote the use of clean agricultural products and the development of green agriculture; (3) strengthen the guidance of the flow of financial funds to support agriculture, channel the flow of monetary funds to modern and low-carbon green agricultural production, increase investment in agricultural production infrastructure, and promote the green development of agriculture; and (4) improve the research and development of agricultural green technology, establish a green agricultural technology innovation alliance, facilitate the opening and sharing of green agricultural production resources and the construction of a service platform base, and improve the efficiency of agricultural production. The trend of China’s transformation to green development brooks no delay, and green agricultural production is an indispensable part of this development. Developing green agriculture still requires the joint efforts of the whole society.

## Figures and Tables

**Figure 1 ijerph-19-09043-f001:**
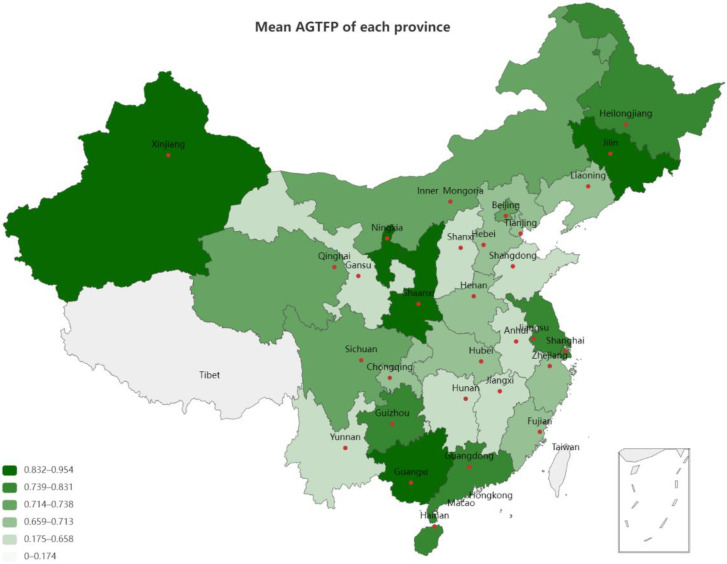
The distribution of agricultural green total factor productivity(AGTFP) in each province.

**Figure 2 ijerph-19-09043-f002:**
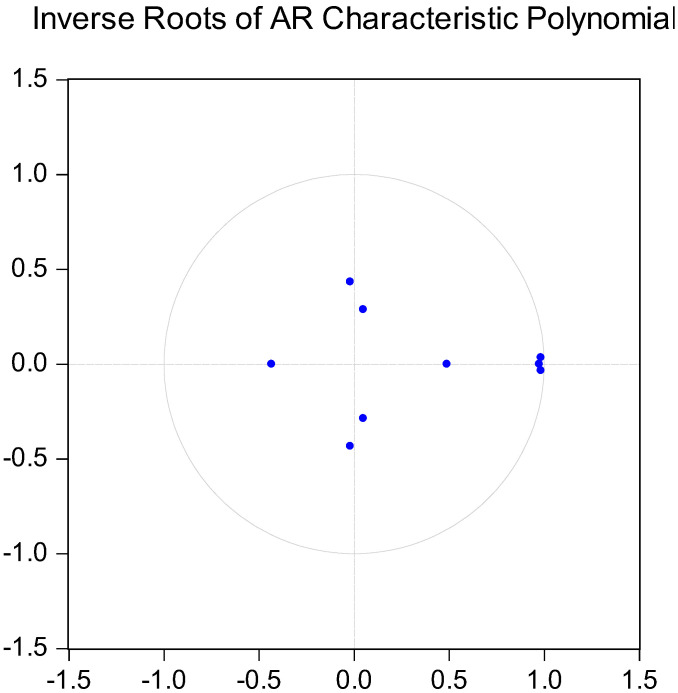
The inverse roots of the AR characteristic polynomial.

**Figure 3 ijerph-19-09043-f003:**
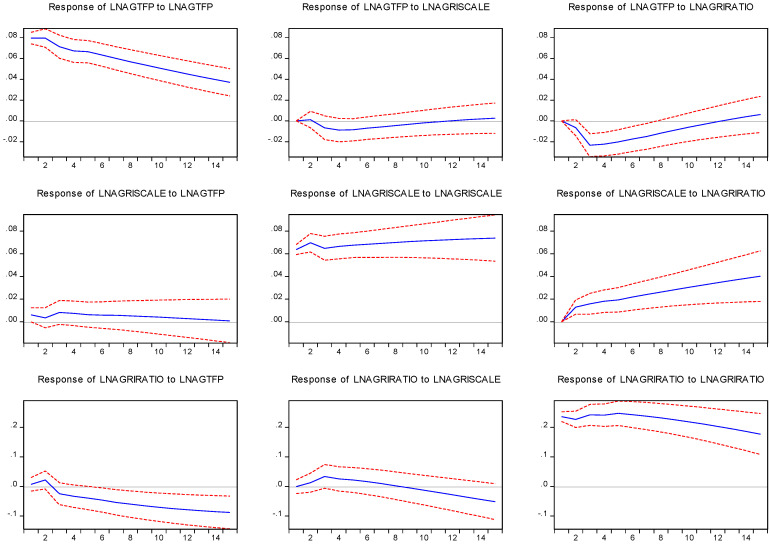
The impulse response of lnAGTFP, lnAGRISCALE, and lnAGRIRTIO for prediction of 15 years (blue color) with a 95% of confidence interval (red color).

**Table 1 ijerph-19-09043-t001:** Carbon emission coefficients of different elements.

Carbon Source	Carbon Emission Coefficient	Reference Sources
Fertilizer	0.8956 kg/kg	Oak Ridge National Laboratory
Pesticide	4.9341 kg/kg	Oak Ridge National Laboratory
Agricultural plastic films	5.18 kg/kg	Institute of Resource, Ecosystem, and Environment of Agriculture, Nanjing Agricultural University
Agricultural diesel oil	0.5927 kg/kg	Intergovernmental Panel on Climate Change (IPCC)
Agricultural cultivation	312.6 kg/hm^2^	College of Biological Sciences, China Agricultural University
Agricultural irrigation	25 kg/hm^2^	[44,45]
Pigs	34.0910 kg/(each·year)	Intergovernmental Panel on Climate Change (IPCC)
Cattle	415.91 kg/(each·year)	Intergovernmental Panel on Climate Change (IPCC)
Sheep	35.1819 kg/(each·year)	Intergovernmental Panel on Climate Change (IPCC)
Agricultural electricity	0.7921 t·MWh^−1^	Ministry of Ecological Environment

**Table 2 ijerph-19-09043-t002:** Descriptive Statistics.

Variable	Mean	Std. Dev.	Min	Max
Fertilizer	175.131	137.603	6.2	716.1
Pesticide	5.266	4.224	0.14	17.35
Agricultural plastic films	6.955	6.33	0.06	34.35
Agricultural diesel oil	63.827	64.681	1.8	487
Agricultural cultivation	5327.101	3588.664	88.6	14783.4
Agricultural irrigation	2004.289	1515.2	109.24	6177.59
Agricultural electricity	212.195	337.304	1.5	1949.1
Cattle	357.893	293.474	1.2	1496.2
pig	1515.626	1282.969	13.2	5757
sheep	973.076	1200.783	11	6111.9
Total carbon emission	630.554	414.268	18.776	1996.382
AGRISCALE	0.602	0.287	0.209	2.618
AGRIRATIO	0.092	0.042	0.012	0.19
AGTFP	0.718	0.166	0.071	1
LnAGRISCALE	−0.593	0.39	−1.566	0.963
LnAGRIRATIO	−2.519	0.57	−4.439	−1.663
lnAGTFP	−0.37	0.317	−2.645	0

**Table 3 ijerph-19-09043-t003:** The results of cross-sectional dependence tests.

Test	Statistic	Prob.
Breusch–Pagan LM	2655.340	0.0000
Pesaran scaled LM	75.27657	0.0000
Pesaran CD	32.25713	0.0000

**Table 4 ijerph-19-09043-t004:** The results of panel unit root tests.

Variables	Level		First-Difference	
	with Constant	Constant and Trend	Constant	Constant and Trend
LLC test				
lnAGRISCALE	0.5205	0.1479	0.0000	0.0000
lnAGRIRATIO	0.0000	0.7712	0.0000	0.0000
lnAGTFP	0.9987	0.0002	0.0000	0.0000
Im, Pesaran, and Shin test				
lnAGRISCALE	1.0000	0.4135	0.0000	0.0000
lnAGRIRATIO	0.0000	1.0000	0.0000	0.0000
lnAGTFP	1.0000	0.1861	0.0000	0.0000
ADF-Fisher Chi-square test				
lnAGRISCALE	0.9991	0.3571	0.0000	0.0000
lnAGRIRATIO	0.0000	1.0000	0.0000	0.0000
lnAGTFP	0.8787	0.0004	0.0000	0.0000
PP-Fisher Chi-square test				
lnAGRISCALE	0.9994	0.6455	0.0000	0.0000
lnAGRIRATIO	0.3108	1.0000	0.0000	0.0000
lnAGTFP	0.9711	0.0033	0.0000	0.0000
Breitung t-stat test				
lnAGRISCALE	-	0.6116	-	0.0000
lnAGRIRATIO		0.6061	-	0.0000
lnAGTFP	-	1.0000	-	0.0000

**Table 5 ijerph-19-09043-t005:** The results of Kao’s residual panel cointegration test.

	Null Hypothesis	t-Statistics	Probability
ADF	No co-integration	−1.779229	0.0376

**Table 6 ijerph-19-09043-t006:** The results of Pairwise Granger Causality Tests.

Null Hypothesis:	F-Statistic	Prob.
H0: LNAGRISCALE does not Granger Cause LNAGTFP	1.12416	0.3462
H0: LNAGTFP does not Granger Cause LNAGRISCALE	1.31997	0.2323
H0: LNRIRITIO does not Granger Cause LNAGTFP	3.68307	0.0004
H0: LNAGTFP does not Granger Cause LNRIRITIO	0.36103	0.9401
H0: LNRIRITIO does not Granger Cause LNAGRISCALE	3.22780	0.0016
H0: LNAGRISCALE does not Granger Cause LNRIRITIO	1.49280	0.1601

**Table 7 ijerph-19-09043-t007:** The results of ARDL.

Variable	Coefficient	Std. Error	t-Statistic	Prob.
	Long Run Equation			
LNAGRISCALE	0.575664	0.035050	16.42392	0.0000
LNAGRIRITIO	−0.034418	0.012390	−2.777914	0.0057
	Short Run Equation			
COINTEQ01	−0.237633	0.069231	−3.432468	0.0007
D(LNAGRISCALE)	−0.146240	0.107445	−1.361067	0.1743
D(LNAGRIRITIO)	0.017363	0.022288	0.779053	0.4364
C	0.029774	0.016726	1.780113	0.0758

**Table 8 ijerph-19-09043-t008:** The results of judging the optimal lag order.

Lag	LogL	LR	FPE	AIC	SC	HQ
1	1069.305	NA	7.13 × 10^−8^	−7.942363	−7.821444 *	7.893791 *
2	1081.106	23.07055	6.98 × 10^−8^	−7.963340	−7.721503	−7.866196
3	1095.621	28.05235	6.70 × 10^−8^ *	−8.004654 *	−7.641899	−7.858938
4	1100.766	9.827781	6.90 × 10^−8^	−7.975779	−7.492105	−7.781490
5	1109.866	17.17724	6.90 × 10^−8^	−7.976527	−7.371934	−7.733666
6	1120.839	20.46507 *	6.80 × 10^−8^	−7.991300	−7.265789	−7.699867
7	1128.659	14.41151	6.86 × 10^−8^	−7.982468	−7.136038	−7.642463
8	1137.519	16.12684	6.87 × 10^−8^	−7.981418	−7.014070	−7.592841

* indicates lag order selected by the criteria.

## Data Availability

Data supporting the conclusions of this article are included within the article. The dataset presented in this study are available on request from the corresponding author.

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
