# Peer review of "Scale of Operation, Financial Support, and Agricultural Green Total Factor Productivity: Evidence from China"

_ijerph, 2022, doi:10.3390/ijerph19159043_

Round 1

Reviewer 1 Report

This is a very interesting paper that examines the dynamic relationships in green agriculture with strong implications to government policies, but very difficult to read and make a recommendation due to the need for extensive editing of the English language and writing style. I will therefore suggest a native English speaker goes through the ms to address these issues first. It will not be fair to accept or reject at this time.

Author Response

We are very grateful to the reviewer for taking the time to read our paper. We are deeply sorry for our poor English expression. We have tried to make our language more fluid and structure more logical.

Reviewer 2 Report

The aim of this research was to investigate the dynamic relationship between operation scale, financial support, and agricultural green total factor productivity (AGTFP) on example of 30 provincial administrative regions in China from 2000 to 2019.

The authors described in detail the data sources with their characteristics, as well as the methodology based on a panel autoregressive model with lagged distribution and agricultural panel data.

Firstly, the SBM model including bad output to measure each province's AGTFP. Secondly, they tested the used variables by unit root, cointegration, Granger causality, etc. Thirdly, the paper makes an empirical study on the relationship between three variables by using the panel data model.

Also on the base of literature overview the paper takes the carbon emissions from agricultural production into account when calculating the index of agricultural production efficiency. They pointed out that carbon emission is the key focus of environmental protection and  are in all aspects of agricultural production, calculating agricultural carbon emissions can better reflect green production efficiency.

The empirical results showed that there are significant cross-sectional, cointegration and long-run relationships between the scale of agricultural activity, financial support for agriculture and AGTFP. Moreover, increasing the intensity of financial support to agriculture does not improve AGTFP but rather the opposite.

Additionally:

·      1.   the results of the Granger panel causality test shows that financial support for agriculture has a unidirectional causal relationship with the scale of agricultural activity and AGTFP, respectively.

·     2.    the results of the impulse response test indicate that reducing part of the financial support to agriculture or increasing the scale of activity can promote AGTFP.

At the same time, the authors emphasized 4 policy recommendation e.c.:

·         1. to increase the conversion rate of agricultural land and promote the agglomeration of agricultural land to form large-scale agricultural management units.

·       2.  to stop or reduce subsidies for agricultural supplies with high carbon and pollution emissions, which can effectively reduce the agricultural scale of non-clean production in order to pursue profits, promote the use of clean agricultural production products, and promote the development of green agriculture.

·        3.  the is to strengthen the guidance of the flow of financial funds to support agriculture, promote the flow of monetary funds to modern, low-carbon green agricultural production, increase investment in agricultural production infrastructure, and promote the green development of agriculture.

·         4. to strengthen the research and development of agricultural green technology, establish a green agricultural technology innovation alliance, promote the opening and sharing of green agricultural production resources and the construction of a service platform base, and improve the efficiency of agricultural production.

These conclusions and recommendation  have long-term practical significance for agricultural departments and financial distribution decision-making

I recommend the article for publication.

General outlines:

The literature review is relevant to the topic. Most of the cited references are from the last five years.  The citations of older publications give a good background for the discussed issues.

The tables and figures presented in the article enrich the discussed issues and support the description of the presented issues.

Please check the list of the references – some of them has different format (No 30 or 32).

Author Response

We are very happy and grateful to the reviewer for reading and approving our paper.

Point1: Please check the list of the references – some of them has different format (No 30 or 32).

Response1:Thanks for the reviewer's careful observation. The different reason may be that they are books, which are automatically generated by Mendeley according to the requirements of the periodical format

Author Response

First of all, we would like to thank Reviewer 3 for reading our article and for your valuable comments. We have marked the revisions in yellow. Below is our response to all comments made.

Point1:the authors should convincingly explain why in their research the use of the area of agricultural sown area per capita is a better measure than farms land size , for measuring agricultural operation scale

Response1: We thank the reviewer for raising this question. We have added in the paper why in their research the use of the area of agricultural sown area per capita is a better measure than farms land size. Generally speaking, from a micro point of view, the agricultural operation scale is expressed by the land size of each farm [1–4]; from a macro perspective, the scale of agricultural operation is expressed by the per capita cultivated land area of each region or the sown area of each agricultural laborer [1,3]. Because of the historical reasons of agricultural production, China is still dominated by small-scale peasant production in the past 20 years, but it is also transforming to large-scale farm production[5,6]. Based on the above practical reasons, it is better to explore the impact of agricultural operation scale on AGTFP from a macro perspective than to use micro farm data

  1. Hu, Y.; Li, B.; Zhang, Z.; Wang, J. Farm Size and Agricultural Technology Progress: Evidence from China. Journal of Rural Studies 2019.
  2. Guth, M.; StÄ™pieÅ„, S.; SmÄ™dzik-Ambroży, K.; Matuszczak, A. Is Small Beautiful? Techinical Efficiency and Environmental Sustainability of Small-Scale Family Farms under the Conditions of Agricultural Policy Support. Journal of Rural Studies 2022, 89, 235–247.
  3. Zhu, W.; Wang, R. Impact of Farm Size on Intensity of Pesticide Use: Evidence from China. Science of the Total Environment 2021, 753, 141696.
  4. Debonne, N.; van Vliet, J.; Ramkat, R.; Snelder, D.; Verburg, P. Farm Scale as a Driver of Agricultural Development in the Kenyan Rift Valley. Agricultural Systems 2021, 186, 102943.
  5. Zhang, Q.; Chu, Y.; Xue, Y.; Ying, H.; Chen, X.; Zhao, Y.; Ma, W.; Ma, L.; Zhang, J.; Yin, Y.; et al. Outlook of China’s Agriculture Transforming from Smallholder Operation to Sustainable Production. Global Food Security 2020, 26, 100444.
  6. Rogers, S.; Wilmsen, B.; Han, X.; Wang, Z.J.H.; Duan, Y.; He, J.; Li, J.; Lin, W.; Wong, C. Scaling up Agriculture? The Dynamics of Land Transfer in Inland China. World Development 2021, 146, 105563.

Point2: measure of financial support for agriculture (the ratio of agriculture - related expenditure to the total financial expenditure ) does not seem appropriate, because it depends on expenditure that has nothing to do with the AGTFP. A better solution would be to use a measure illustrating the level of expenditure for agriculture in relation to the area of agricultural crops)

Response2: We are very grateful to the reviewer for the suggestion on the measurement indicator. The index for measuring fiscal expenditure put forward by the reviewer is excellent. However, because the agricultural operation scale, another very important variable in this paper, already includes the area of agricultural crops, to avoid the high correlation between variables and to increase the information provided by variables, this paper uses a financial relative expenditure index. Using the relative indicator of fiscal expenditure has the following advantages: on the one hand, it can avoid the impact of the rapid increase in China's economy over the past 20 years. On the other hand, policy makers can well control this relative indicator due to the high degree of centralization of power by the Chinese government.

Point3: it is not clear why financial support for agriculture has a negative impact on AGTFP, and the assumption that such support leads to the intensification of agricultural production and consequently harms the environment is a far-reaching simplification, because farmers can be rewarded for reducing the intensity and taking greater care for the environment, as is the case, for example, in the European Union

Response3: I am very sorry that our presumption of the conclusion puzzled the reviewer. Different from European agricultural production, because of less arable land, large population, large demand for food, and in the stage of rapid economic development, the Ministry of Agriculture does not pay enough attention to environmental protection in India and East Asia. So subsidies for output will cause farmers to overproduce and damage the environment. In addition, we added the following sentence in the discussion paragraph: “This paper mainly discusses the influence of the intensity of financial support for agriculture on AGTFP from a macro point of view, but this study still has some limitations such as ignoring the role of specific ways of financial support for agriculture, and it needs more works to do to explain the overall negative impact of the intensity of financial support for agriculture”

Point4: there is no discussion in the manuscript with the results of other authors dealing with the topic discussed in the article. The authors should explain whether only in Chinese agriculture increasing the scale of agricultural activity and reducing financial support has a positive effect on AGTFP, or there are similar results from research conducted in other countries. It would also be useful for the readers to explain - in the literature review - the similarities and differences between AGTFP and the concept of sustainable agriculture (as it is viewed by the authors).

Response4:

  • I am very grateful to the reviewer for the suggestion on the insufficient extension of our conclusions. We have added a discussion section, as follows:

Due to the different research perspectives and the choice of indicators to measure the high-quality development of agriculture, previous studies obtained results which are different from that of this paper. Although the empirical test results of this paper indicate that the overall impact of the intensity of financial support for agriculture on AGTFP is negative, it may have a positive effect on AGTFP when specific to a certain type of specific fiscal expenditure. For example, Deng et al. (2021) found that public agricultural R&D investment is the main driver of China's AGTFP improvement [7]; in Malawi, government subsidies for production factors to improve the efficiency of small-scale agricultural production are conducive to agricultural green development [8]. Moreover, within the provincial administrative regions, the impact of agricultural operation scale on AGTFP is negative. Hu et al. (2019) found that the smaller the scale of agricultural operation, the higher the utilization rate of input factors in Jiangsu, China [9].

(2) According to the suggestion of the reviewer, we added the following sentence to the review. “The sustainable development of agriculture can be simply summarized as meeting the needs of contemporary people for agricultural products without harming the ability of future generations to develop agriculture following the Brundtland Report (1987). However, due to the different evaluation systems, the interpretation of agricultural sustainable development is vague and complex [10,11].”

  1. Deng, H.; Jin, Y.; Pray, C.; Hu, R.; Xia, E.; Meng, H. Impact of Public Research and Development and Extension on Agricultural Productivity in China from 1990 to 2013. China Economic Review 2021, 70, 101699.
  2. Abman, R.; Carney, C. Agricultural Productivity and Deforestation: Evidence from Input Subsidies and Ethnic Favoritism in Malawi. Journal of Environmental Economics and Management 2020, 103, 102342.
  3. HU, L. xiao; ZHANG, X. heng; ZHOU, Y. heng Farm Size and Fertilizer Sustainable Use: An Empirical Study in Jiangsu, China. Journal of Integrative Agriculture 2019, 18, 2898–2909.
  4. Laurett, R.; Paço, A.; Mainardes, E.W. Sustainable Development in Agriculture and Its Antecedents, Barriers and Consequences – An Exploratory Study. Sustainable Production and Consumption 2021, 27, 298–311.
  5. Laurett, R.; Paço, A.; Mainardes, E.W. Antecedents and Consequences of Sustainable Development in Agriculture and the Moderator Role of the Barriers: Proposal and Test of a Structural Model. Journal of Rural Studies 2021, 86, 270–281.